# Physiological Stress Responses in Cattle Used in the Spanish Rodeo

**DOI:** 10.3390/ani13162654

**Published:** 2023-08-17

**Authors:** Sara Caceres, Julia Moreno, Belen Crespo, Gema Silvan, Juan Carlos Illera

**Affiliations:** Department of Animal Physiology, Veterinary Medicine School, Complutense University of Madrid (UCM), 28040 Madrid, Spain; sacacere@ucm.es (S.C.); juliamorenodetejada@gmail.com (J.M.); belencre@ucm.es (B.C.); jcillera@ucm.es (J.C.I.)

**Keywords:** catecholamines, cortisol, dopamine, serotonin, calves, animal welfare

## Abstract

**Simple Summary:**

There are different public events that cause a distress response in livestock. Persecution and takedown is a sport similar to rodeo that can affect the animals that compete in it. Therefore, in this study, the physiological stress response is evaluated in saliva samples from calves before, during, and after the event, measuring the levels of epinephrine, cortisol, serotonin, and dopamine. The results revealed that the sport of persecution and takedown could cause a punctual acute stressful effect that can be assumed by animals and not compromise animal welfare.

**Abstract:**

Certain events can cause distress in cattle. In Spain, there is a sport similar to rodeo called persecution and takedown, in which calves are harassed and knocked down by riders. In this study, the physiological stress response of calves (n = 260) is assessed by measuring hormonal physiological parameters. Salivary samples were collected from Salers (n = 110) and Lidia (n = 150) calves before, during, and after the persecution and takedown event. The hormones epinephrine, cortisol, serotonin, and dopamine were determined in saliva samples using enzyme-immunoassay techniques. The results obtained revealed that epinephrine and cortisol levels increased during the event in Salers calves, with a significant increase (*p* < 0.05) in the case of epinephrine, although after the event, these values returned to their initial state. Therefore, this sport supposes an assumable punctual stressor stimulus for the animal. In contrast, in Lidia calves, cortisol and epinephrine levels decreased, with a significant decrease (*p* < 0.05) in the case of cortisol, which may be related to the temperament of this breed and facing a stressful situation in a different manner. This is confirmed by serotonin and dopamine levels that were altered in Lidia calves with respect to the other group studied. In conclusion, the sport of persecution and takedown produces a physiological response of adaptive stress assumable for the animals.

## 1. Introduction

Cattle are important in the meat industry, but they are also part of various public events around the world. In North America and Australia, animals such as calves are used in rodeos [1]. In these events, calves are harassed and roped by riders, which can be considered a stressor for the animal [1]. Similarly, in Spain, there is a competitive sport called “acoso y derribo” (persecution and takedown), in which calves, in their natural habitat, are harassed and knocked down by riders.

Persecution and takedown is an equestrian competition. It consists of a team of two participants on horseback leading a cattle animal within an enclosed area, in a certain time, to subsequently knock it down, thus being able to assess its character and manageability. This type of practice is only carried out once in the life of the animals [2].

This sporting activity has its origins in a handling and testing procedure used by bull breeders from the nineteenth century. It consisted of chasing the cattle from horses and knocking them down with a pole to observe how they behaved. Although in its beginnings, it was a livestock practice, currently, the Spanish Equestrian Federation has reinstituted it as a competitive sport [2].

The training of the persecution and takedown discipline, like any other equestrian discipline, requires the preparation of the horses. For this purpose, it is necessary to practice persecution and takedown outside of competition in enclosures designed for this purpose that meet the requirements of the discipline. This training involves the use of bovine animals of different breeds, ages, and weights; the health condition of the cattle must be optimal for the proper development of the training, and the health of the animal must always be ensured without exceeding the number of runs [2].

The scheme of the route taken by these animals is shown in Figure 1. The animals are kept in the accustoming corrals for two days prior to the competition. At the time of the competition, they are led to the competition corral that starts in the rodeo area. From this area, they are released and harassed by a pair of riders through the corridor, which is about 600 m long. During the race, the animals reach a speed of 20–30 km/h [1,3], which is an important physiological effort. When passing through the area known as the quadrilateral, which is delimited by flags and is about 100 m long, the horseman must knock down the calf. This is normally performed by a light touch of the pole vault in the rump area, which is the highest part of the calf around the root of the tail [3]. The animals can be knocked down a total of three times in the case of Salers calves (due to their tamer attitude) and only once in the case of Lidia calves, which are characterized by their bravery. Depending on the knockdown, the rodeo judges will quantify the conditions in which it has occurred and will qualify the acting pair of riders. The score is obtained according to how the fall of each of the cattle is performed. For each tumble, the couple scores six points, and three points for each throw. If the rider catches the animal and releases it and if he passes the pole over it, three points are deducted [2].

Animal–human interaction varies the animal’s behavioral and physiological response resulting, in some cases, in a stressful stimulus for the animal. Depending on the magnitude and type of the stressful stimuli, animals can modulate their physiological response [4].

There are studies on animal welfare in fighting cattle farms where standardized indicators of direct observation are analyzed [5]. Hormones such as glucocorticoids or epinephrine have been used as indicators of stress in bovine species [1]. Physical stress stimulates the hypothalamus–pituitary–adrenal (HPA) and sympathetic nervous systems. Cortisol has various physiologic effects, including catecholamine release, suppression of insulin, mobilization of energy stores through gluconeogenesis and glycogenolysis, suppression of the immune-inflammatory response, and delayed wound healing [6]. Any physical or psychological stimuli that disrupt homeostasis result in a stress response. The stimuli are called stressors, and physiological and behavioral changes in response to exposure to stressors constitute the stress response. A stress response is mediated by a complex interplay of nervous, endocrine, and immune mechanisms that involves activation of the sympathetic-adreno-medullar (SAM) axis, the HPA axis, and the immune system [6,7].

This stress response is associated with aspects of the animal’s behavior, including the signaling pathway of serotonin and dopamine. These signaling pathways have been considered to harbour genetic variations that may be associated with variable behavioral responses and are central to behavioral phenotypes in cattle. Nevertheless, alterations in the noradrenergic system have been associated with cognitive and neurological disorders related to abnormal social behavior [8].

In the persecution and takedown competition, animals are subjected to a large number of stimuli that can give rise to different physiological responses. And parameters such as cortisol, epinephrine, serotonin, and dopamine can be altered during competition. Therefore, the aim of this study was to assess whether there are variations in the physiological hormonal parameters used as a measure of physiological stress in persecution and takedown calves. Thus, the levels of cortisol, epinephrine, dopamine, and serotonin in saliva samples collected from calves in different situations related to this activity were analyzed to assess the response of the animals to a stressful situation, as well as their ability to recover.

## 2. Materials and Methods

### 2.1. Animals

A total of 260 animals, females between 8 and 10 months of age, were sampled. The average body weight was approximately 180–250 kg. The breeds of the animals analyzed were the following.

Lidia (n = 150): The Lidia cattle breed originated in Spain during the Middle Ages and is characterized mainly by its rusticity and behavior. Its breeding system is completely extensive, which has an important effect on landscape conservation [9]. It is intended for corrida shows.

Salers (n = 110): This breed is a French native and dual-purpose (beef/dairy), which originates from the high Massif Central region of France. It is a rustic breed used for beef production and shows excellent maternal aptitudes, hardiness, and satisfactory milk yield [10].

Saliva samples from these animals were collected by inserting polypropylene-cotton swabs (cotton tip Ø11 mm) for a few seconds under the tongue and between the tongue and cheeks, then cutting the tips of the swabs and placing them in 5 mL cryotubes. The weight of the cotton tips is 0.5 g, and usually, between 0.5 and 1 mL of saliva was obtained after centrifugation. Samples were collected at different times: before, during, and after the event, obtaining three samples from each animal. Before the event, a sample is collected from the animal in the accustoming corral 15 min before its participation in the contest in a cattle chute. During the event, it is collected immediately after the knockdowns. After the event, it is collected 2 h after the competition in a cattle chute [2]. All the samples were collected between 11:00 a.m. and 1:00 p.m. to avoid possible circadian variations. All the shows where the samples were collected were located in the province of Badajoz (Spain) to avoid changes due to latitude during the spring.

All the procedures involving animals were reviewed and approved by the Animal Research and Ethics Committee of the Complutense University of Madrid (Reference number: UCM 0018/003).

### 2.2. Sample Processing

A total of 780 samples were collected. All hormones were measured at 48 h after their collection. Saliva samples were kept refrigerated at 4 °C from collection to processing. Swabs were centrifuged at 1200× *g* for 20 min at 20 °C in order to obtain the saliva sample. Then, saliva samples were stored in eppendorfs and frozen at −20 °C until hormonal analysis.

### 2.3. Hormonal Analysis

The hormones analyzed in the saliva samples were cortisol, epinephrine, dopamine, and serotonin. A competition enzyme-immunoassay (EIA) technique was used for the determination of cortisol in saliva. This technique has been validated by the Endocrinology Laboratory of the Departmental Section of Veterinary Physiology, Complutense University of Madrid [11].

Briefly, 96-well flat-bottomed polystyrene microplates (Biohit, Finland) were coated overnight at 4 °C with the dilution of anti-cortisol antibody (1/8000). After washing the plates, standards and saliva samples diluted in the working solution of cortisol conjugate (1/80,000) were added to the plate in duplicate and incubated for 2 h. To determine the amount of salivary cortisol, Enhance K-Blue TMB substrate (Neogen, Lexington, KY, USA) was added, and the reaction was stopped by the addition of 10% H_2_SO_4_. The absorbance was read at 450 nm using a SpectraMax 190UV/Vis 96-well automated plate reader. Hormone concentrations were calculated using software developed for this technique (ELISA AID, Eurogenetics, Brussels, Belgium).

Epinephrine, dopamine, and serotonin analyses were performed using commercial kits according to manufacturer’s instructions (Demeditec Diagnostics GmbH, Kiel, Germany). All commercial kits used are validated for bovine species [12]. All hormone concentrations were expressed in ng/mL.

### 2.4. Statistical Analysis

Analysis was performed using IBM SPSS Statistic 25 software (University Complutense of Madrid). The results were expressed as the means ± SE. The Kolmogorov–Smirnoff test was used to assess the goodness-of-fit distribution of the collected data. Variables studied were noted to be parametric, and an analysis of variance (ANOVA) was performed, followed by Bonferroni post hoc test for the comparison between samples (samples 0, 1, 2) and between groups (Salers and Lidia calves). In all statistical analyses, the confidence level was 95%, and statistically significant differences were considered for *p*-values < 0.05.

## 3. Results

### 3.1. Epinephrine

The results obtained for epinephrine determinations in the saliva of persecution and takedown cattle are shown in Figure 2. In all cases, the measured values of epinephrine in saliva were within the values of the physiological range of the species, between 0 and 10 ng/mL [12,13,14,15].

Significant differences were found in epinephrine values at the time of takedown (sample 1) in the case of Salers calves (*p* < 0.05). However, no significant differences were found among the three samples studied in the group of Lidia calves.

Interestingly, in the Lidia calves group, there was a reversal in the values found since the values measured at the time before the start of the competition were higher, but not significantly, than the values found after the persecution and 2 h after the contest.

On the other hand, comparing the levels of epinephrine in the three samples between groups, we observed that the group of Lidia calves had significantly lower levels of epinephrine in the three samples studied than those found in the Salers calves (*p* < 0.05).

### 3.2. Cortisol

The results obtained for cortisol in the saliva of the persecution and takedown cattle are shown in Figure 3. In all cases, the measured values of cortisol in saliva were within the physiological range of the species, between 0 and 20 ng/mL [16,17,18].

As no significant differences between the three samples were found in the Salers calves group, in the case of the Lidia calves, we observed a significant decrease (*p* < 0.05) in cortisol concentrations at the time of takedown (sample 1) and at 2 h after the end of the activity (Sample 2) with respect to initial intake. 

Indeed, no significant differences were found between Salers and Lidia calves before, during, and after the activity.

#### Serotonin

The results obtained for serotonin in persecution and takedown cattle are shown in Figure 4. 

The values of serotonin in the saliva of the study animals are within the physiological range of the species (0–1000 ng/mL) [19,20,21,22]. In addition, we did not find significant differences in serotonin values at the different times of sampling within each group. However, comparing the serotonin levels among the different groups, we observed that the brave calves group presented significantly lower serotonin levels with respect to the other studied group in all samples (*p* < 0.05).

## 4. Dopamine

The results obtained for dopamine in the saliva of the persecution and takedown cattle are shown in Figure 5.

The dopamine values found are within the physiological range of the species (0–80 ng/mL) in all groups [23,24,25].

Similar to what was found in serotonin levels, dopamine levels also had no significant differences between study samples. Furthermore, in the comparison between groups, we found that, in the case of the Lidia calves, dopamine secretion is significantly higher than in the other study group in all samples studied.

## 5. Discussion

The use of biological indicators to assess physiological stress response is the order of the day. Hormonal indicators such as cortisol or epinephrine are used to assess stress responses in animals [26] in different biological fluids such as blood, milk, saliva, or even feces [27]. However, the physiological stress response is compromised in certain events, such as rodeo, altering these hormone levels [3]. In this study, we wanted to assess the physiological stress response in calves participating in the Spanish sport persecution and takedown by determining cortisol, epinephrine, serotonin, and dopamine levels in saliva samples taken before, during, and after the contest in order to determine the physiological response of these animals.

The choice to use saliva samples was based on the fact that their collection is a non-invasive sampling system with the advantages that this entails for the handling of the animals and, also, the hormones circulate freely and are not bound to proteins [28,29]. Likewise, the additional stressor of needle puncture on the animal is avoided [30] and minimizes the stress caused by the restriction of the animal for blood collection [31]. In addition, it has been observed that the half-life of epinephrine in saliva is much longer than in blood, which favors finding the hormone at longer collection times [19]. This fact provides greater flexibility in sample collection, which in itself is complicated in bovine animals.

It is known that in response to stress situations, the hypothalamic pituitary adrenal axis and sympathetic adrenal medullary axis are activated in order to provide a metabolic response that generally turns into the secretion of different hormones [32]. Thus, the examination of these biomarkers provides valuable information on the status of the animal under physical overexertion and/or stress. Cortisol is the primary effector molecule of the hypothalamic–pituitary–adrenal (HPA) axis involved in response to stressors.

In animals, to determine when a stimulus can provoke a stress response, we use hormonal indicators, such as the concentrations of cortisol or epinephrine that are released in response to these stimuli [4]. In this study, we found that all the concentrations measured at different times of the different hormones analyzed are within the normal physiological ranges of these animals [12,16,33]. Therefore, the variations found are not indicative of acute stress situations but of a punctual stimulus that activates the physiological systems of adaptation to a high-intensity exercise so that the animal responds efficiently to this new situation.

Hormones, such as cortisol and catecholamines, are released by the body in response to stress by stimulating glycogen mobilization and muscle metabolism [26]. In these situations, the hypothalamic–pituitary–adrenal axis is activated, which produces two main events: the release of cortisol by the adrenal cortex and the release of epinephrine by the adrenal medulla, among other mechanisms [34].

In our study, we observed a significant increase in epinephrine concentrations in Salers calves when they were knocked down; however, no significant differences in the other hormones studied were found between samples in this group. This could be to the fact that these types of animals are fattening animals not accustomed to intense exercise. When they go through the persecution and takedown, they are exercised more than usual, and this exercise causes an increase in the release of epinephrine by the adrenal medulla. Nevertheless, the increase found in epinephrine levels in the Salers group during the event is related to an acute and punctual stress situation. Acute stress has been defined as a stimulus that occurs in a short-lived situation that allows a quick adaptation to recover the physiological balance [35]. Therefore, the increase found corresponds to a normal physiological response to a stressor stimulus due to epinephrine levels decreasing after the competition to values similar to those obtained before the competition (before the event). It has been described that the activation of the release of catecholamines makes an animal more capable of coping with stressful situations. These hormones have been described as coping hormones, as they provide energy to the brain in these situations [36].

In the assessment of acute stress, the activation of the hypothalamic–pituitary–adrenal axis led to an increase in circulating cortisol levels in blood, and its quantification is considered the gold standard for assessing stress [37]. However, this study revealed that other hormones, such as epinephrine, can provide additional information on an acute stress response.

Regarding Lidia calves, a different physiological response was found. Higher levels of epinephrine and cortisol were found at the initial time prior to exercise compared to the other two samples studied. This may be due to the Lidia breed being characterized by its natural aggressiveness and resistance to traditional handling procedures. The performance of the Lidia bull during a corrida could be compared to that of an athletic animal due to the intense exercise performed in an unfamiliar and highly stressful environment [32,38]. Therefore, based on hormonal determinations, participating in the competition does not represent a stressful situation for them. Similar to our results, other authors have found an increase in cortisol concentrations during corrida due to the animal’s exercise [39] or increases in epinephrine levels in rodeo animals [3]. However, other authors indicate that there are no statistically significant differences in cortisol levels in lactated and lying animals in the rodeo [40].

This distinct physiological response of the Lidia breed to a stressful stimulus has been previously described [39]. Catecholamines have been described to be involved in the behavior of distant animal species. More docile species showed high levels of catecholamines associated with more aggressive behavior species [26,34,36]. In this study, lower levels of epinephrine and cortisol are observed in the Lidia breed with respect to the other breeds studied at the time of the contest. This change in the levels of these coping hormones may explain the temperament of these animals and how they face this situation differently from the other breeds studied. The selection of the Lidia breed is based on its agonistic–aggressive behavior through a series of traits that classify its aggressiveness and fighting capacity. Cattle temperament is defined as the reactivity to humans and novel environments, and it is associated with stress situations that lead to an animal response [35]. In addition, these traits show significant heritability and thus can be considered suitable for genetic selection [41]. However, in other cattle breeds, aggressiveness can be considered an undesirable trait; it is likely that the selection process for aggressiveness in the Lidia breed has left genomic signatures [42].

This breed-specific temperament is also influenced by the expression of genes related to serotonin and dopamine [8]. Serotonin is a hormone involved in a multitude of processes, such as inhibition of aggression, regulation of body temperature, sleep, sexuality, and control of impulsivity [43]. On the other hand, dopamine is a catecholamine that is secreted in stressful situations, which helps to guide animals away from dangerous objects or environments [34]. It has been found that serotonin and dopamine concentrations do not vary throughout the life of the animal in the bovine species, although differences between breeds can be found [12,19].

In the case of serotonin, a slight non-significant decrease in serotonin was observed at the time of knockdown in Salers calves. This result is in line with other authors who found a negative serotonin–cortisol correlation [21], which agrees with the results of this study, that when an increase in cortisol levels was found, at the same time, a decrease in serotonin levels was also found.

Regarding Lidia calves, we found significant differences in the levels of both hormones with respect to Salers calves. Serotonin levels were lower with respect to the other group, while dopamine levels were higher.

These differences in the concentrations of these hormones may be due to the fact that the Lidia breed has special characteristics since they are more aggressive animals than the other breeds studied. There is scientific evidence that points to the involvement of serotonin in the adaptation of animals to different situations. It has been shown that tryptophan-free diets increase the level of aggressiveness and that low serotonin levels are related to antisocial behavior and violence [3]. This agrees with our results since the breed with lower serotonin levels is the Lidia breed. Therefore, taking these results together, we could suggest that higher levels of dopamine and lower levels of serotonin promote a less docile temperament in these animals that provides them an adaptive response to stressful situations that is different from other breeds. Therefore, these results indicate that these animals present a greater response to stress and a greater capacity to adapt to stressful stimuli, as already described by other authors for the Lidia breed [12,19].

The practical applications of this study are that in the cattle breeds studied, the measurement of the hormones cortisol, epinephrine, serotonin, and dopamine before a specific stressful stimulus provides us with information about the animal’s physiological response to this stimulus.

## 6. Conclusions

Sports such as persecution and takedown may cause distress to the species that participate in them. However, our results support that the variations found in the levels of coping hormones (cortisol, epinephrine, dopamine, and serotonin) are within the physiological range of these animals. Moreover, the variations found at the time of the knockdown represent a timely physiological response to adaptive stress that is perfectly acceptable to the animal.

## Figures and Tables

**Figure 1 animals-13-02654-f001:**
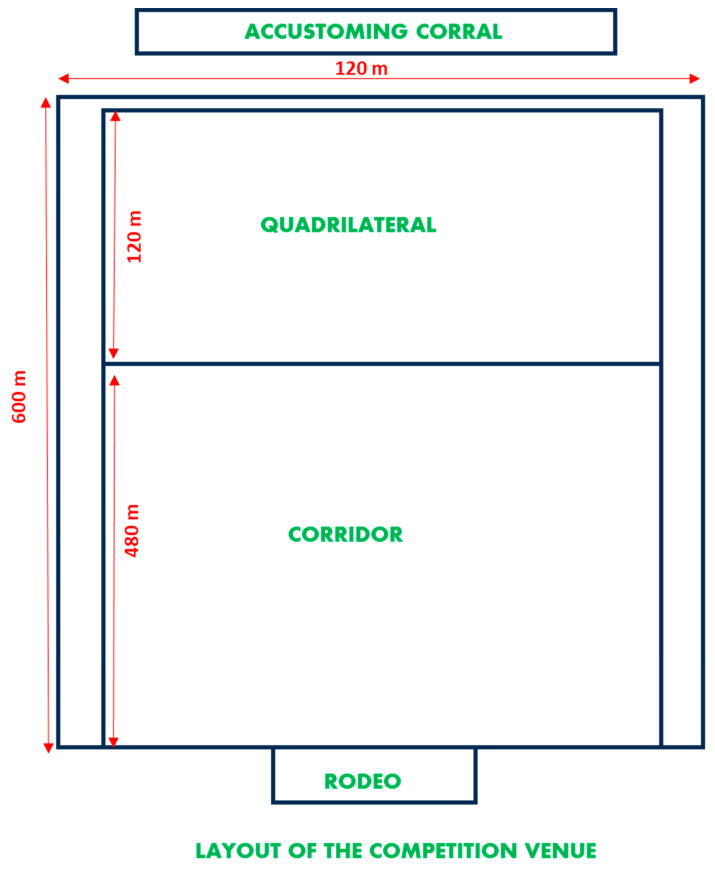
Scheme of the persecution and takedown enclosure: Rodeo = rodeo: from where the animals start. Corredero = corridor: where the animals are harassed. Cuadrilatero = quadrilateral: where the animals are knocked down. Corrales de querencia = accustoming corral: where the animals are kept before the competition [2].

**Figure 2 animals-13-02654-f002:**
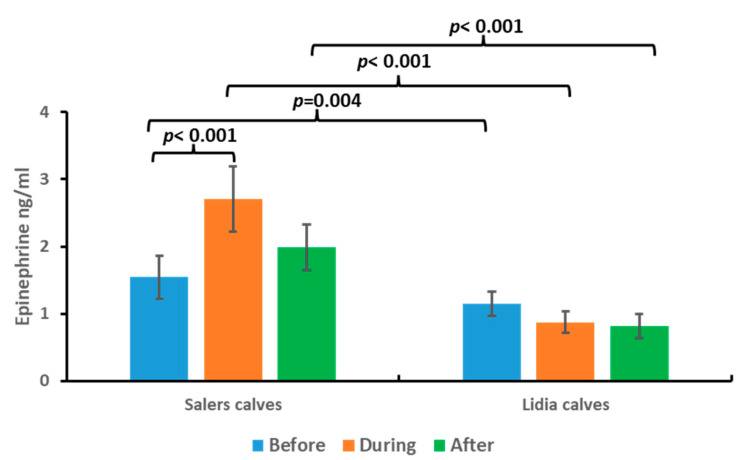
Epinephrine concentrations before, during, and after persecution and takedown competition in Salers calves and Lidia calves. Bars represent media ± SD. Brackets represent significant differences between before and during the event involving Salers calves and significant differences between Salers and Lidia calves on each sample.

**Figure 3 animals-13-02654-f003:**
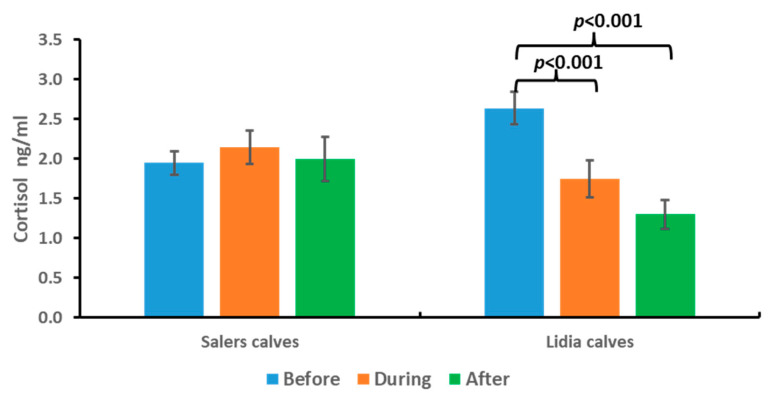
Cortisol concentrations before, during, and after persecution and takedown competition in Salers calves and Lidia calves. Bars represent media ± SD. Brackets represent significant differences between before and during the event and before and after the event in the Lidia calves group.

**Figure 4 animals-13-02654-f004:**
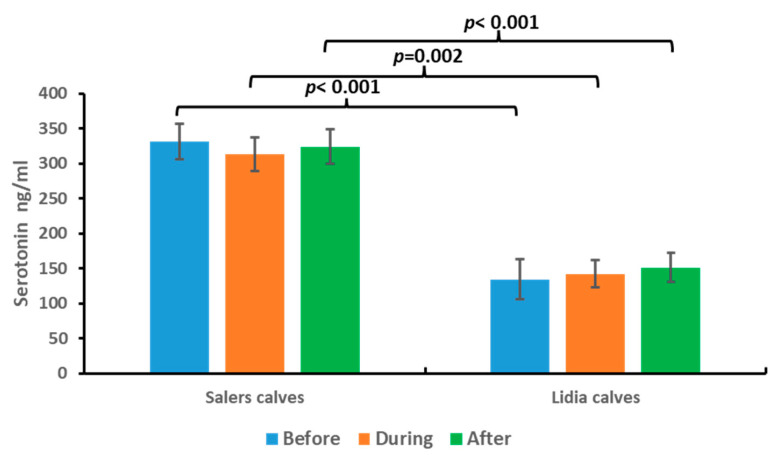
Serotonin concentrations before, during, and after persecution and takedown competition in Salers calves and Lidia calves. Bars represent media ± SD. Brackets represent significant differences between Salers and Lidia calves at each moment of the event.

**Figure 5 animals-13-02654-f005:**
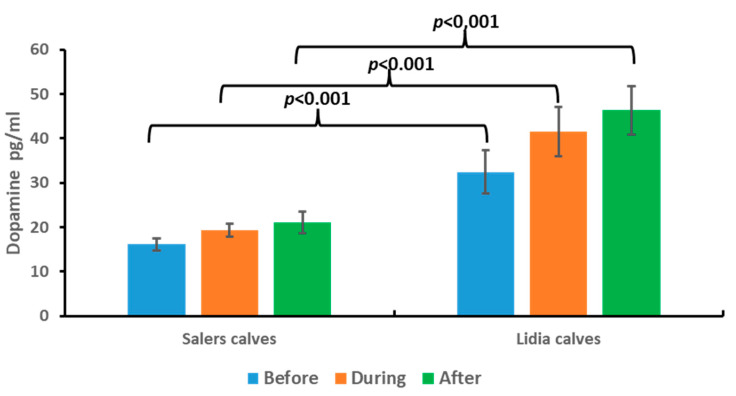
Dopamine concentrations before, during, and after persecution and takedown competition in Salers calves and Lidia calves. Bars represent media ± SD. Brackets represent significant differences between breeds at each moment of the event.

## Data Availability

The data that support the findings of this study are available from the corresponding author upon reasonable request.

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
