# Peer review of "Physiological Stress Responses in Cattle Used in the Spanish Rodeo"

_animals, 2023, doi:10.3390/ani13162654_

Round 1
Reviewer 1 Report
Dear Authors and Editor,
I expose the errors found and questions to improve in the article:
I think that the introduction should describe in a complete way what the sporting event consists of, its origin, what types of animals participate, the rules, if it is a sport or a livestock practice, etc.
P2 – 51: explain how they have measured the speed of the calf in competition.
P1 – 59: An important and repeated error at work, especially in the discussion, is the idea that calves are trained on farms with horses running through the pasture. This is not true, if it is true that adult bulls, from 3-4-5 years old, are trained with running tracks to provide them with physical shape for the bullfight. In no case are calves trained, much less females, as is the case with the sampled animals. The authors will have to find a new argument to justify the results.
P3 – 86: the authors speak of meek and brave breeds. It is necessary to know what races the two groups are made up of since the data could be interesting. There are many differences in tame breeds, between dairy and meat there is a lot of genetic, ethological and physiological difference. and in the brave races? is there more than one??? As far as I know there are several breeds used in bullfighting in Europe: lidia breed, betizu breed and camarguesa breed. The authors must specify which breeds are involved and if they are pure or crossed.
If the animals purely belong to only two breeds: the Saler breed and the Lidia breed, talk about these two breeds in each group.
What was the age of the calves? His weight?? why were they all female? males are not used in the test?
P3 – 91 brave calves n= 1500 ???? it must be a mistake
correct the “:”
How is the saliva sample taken? Was the animal immobilized by any method? how many people? describe the sample collection material.
we understand that animals will not let themselves put a swab in their mouth and will need to immobilize them
Was it done in the same way in the Saler breed as in the Lidia breed?
It causes a lot of intrigue as to how sample number 1 is collected, during the test, did all the animals make the same physical effort before taking the sample? How long did the harassment and demolition exercise last? were they shot down only once? how far did they travel?
Describe the figure 1 with measures of length
P4 -140: Can it be said that the animal, with all the hormones analyzed in physiological ranges, suffers stress? justify the answer.
P5 – 146: provide the normality test and ANOVA values, in each case that significant differences have been found: F -value and p -value, and also identify which values show the differences.
P6 – 174-177: It is not clear from the graphs where the significant differences are.
P6 – 187: when talking about physiological ranges, only one article was referenced, they should look to see if there is a consensus on the ranges because they are sometimes very variable. It would be interesting to introduce more references of accepted physiological ranges.
P7- 202: end point missing.
P7 – 206: It has been shown that the stress in cattle is more due to their immobilization for sample taking than to the puncture itself, change the argument.
P7- 239: As I have said, it is not true that calves are subjected to exercise on farms. justify the results from a genetic point of view. For this they have many articles to review on genetics and behavior of the lidia breed.
P8 – 243: review more bibliography.
P8 – 252: explain and describe what the temperament of these animals is like, reviewing scientific papers that corroborate it.
P8 – 265: the authors say that there is greater stress at some point, how have they measured it?
The authors speak of a correlation in the concentrations of the hormones but no correlation analysis has been done, it would be interesting and obligatory to do so if they want to maintain the affirmations.
P8- 283: Is the test a sport or a livestock practice?
It is necessary and interesting to discuss the different concentrations found in the different races, because these differences exist and to talk about how the sex variable can influence them. For all this, it is necessary to carry out a good bibliographical review on the subject since there are several publications on blood values in lidia cattle that the authors have not reviewed, they only refer to some thesis of the authors' research group.
Author Response
Dear Authors and Editor,
I expose the errors found and questions to improve in the article:
Thank you for all your suggestions that have helped to improve the quality of the manuscript.
I think that the introduction should describe in a complete way what the sporting event consists of, its origin, what types of animals participate, the rules, if it is a sport or a livestock practice, etc.
Thank you very much for your feedback, we have taken it into account and have introduced aspects of the harassment and demolition in the introduction section.
Any breed of cattle can participate in this type of competition, although preferably breeds that are linked to bullfighting are used, for this reason females are used since the males that are going to be fought cannot participate in the competition.
We have introduced the following information
This “sporting" activity its origins in a handling and testing procedure used by bull breeders from the nineteenth century. It consisted of chasing the cattle from horses and knocking them down with a pole to observe how they behaved.
Although in its beginnings it was a livestock practice, nowadays the Spanish Equestrian Federation reinstitutes it as a competitive sport.
The score is obtained according to how the fall of each of the cattle is performed. For each tumble, the couple scores six points and three points for each throw. If the rider catches the animal and releases it and if he passes the pole over it, three points are deducted.
P2 – 51: explain how they have measured the speed of the calf in competition.
The speed of the calf is included in the book of the authors JM Cossio and A Diaz Cañabate, entitled: Los toros: Tratado técnico e histórico.
We have added the reference in the text
P1 – 59: An important and repeated error at work, especially in the discussion, is the idea that calves are trained on farms with horses running through the pasture. This is not true, if it is true that adult bulls, from 3-4-5 years old, are trained with running tracks to provide them with physical shape for the bullfight. In no case are calves trained, much less females, as is the case with the sampled animals. The authors will have to find a new argument to justify the results.
Thank you very much for your suggestion, indeed young females are not moved in the pasture.
We have deleted that sentence in the introduction
P3 – 86: the authors speak of meek and brave breeds. It is necessary to know what races the two groups are made up of since the data could be interesting. There are many differences in tame breeds, between dairy and meat there is a lot of genetic, ethological and physiological difference. and in the brave races? is there more than one??? As far as I know there are several breeds used in bullfighting in Europe: lidia breed, betizu breed and camarguesa breed. The authors must specify which breeds are involved and if they are pure or crossed.
If the animals purely belong to only two breeds: the Saler breed and the Lidia breed, talk about these two breeds in each group.
The two breeds used in the study are pure breeds (within the purity of cattle breeds).
We have changed the name of Brava to "raza de lidia" (fighting breed).
What was the age of the calves? His weight?? why were they all female? males are not used in the test?
The age is entered in material and methods and only females were used because they are the only ones that participated in the contest.
P3 – 91 brave calves n= 1500 ???? it must be a mistake
Thank you very much for your suggestion, it was indeed a mistake since there are 150 animals.
correct the “:”
We have corrected it in the text
How is the saliva sample taken? Was the animal immobilized by any method? how many people? describe the sample collection material.
we understand that animals will not let themselves put a swab in their mouth and will need to immobilize them
The animals in samples 0 and 2 (before and after the competition) were taken by a person introducing the animal in a cattle chute. In the case of the sample 1 (at the time of knocking down the animal) the sample was taken by a person, this sample can be taken because when knocking down the animal the rider keeps the stick on the animal's back and the animal is immobilized and the person taking the sample was in an off-road vehicle to collect it.
Was it done in the same way in the Saler breed as in the Lidia breed?
If it was done in the same way
It causes a lot of intrigue as to how sample number 1 is collected, during the test, did all the animals make the same physical effort before taking the sample? How long did the harassment and demolition exercise last? were they shot down only once? how far did they travel?
As detailed in the introduction section: The animals can be knocked down a total of three times in the case of Salers calves, and only once in the case of brave calves. Depending on the breed, they performed the same physical effort. In the case of the Lidia breed, the animal was only knocked down once, because as soon as they joined the horse, the animal turned back to the horse and it was no longer possible for it to continue running to be knocked down again. This is characteristic of the Lidia breed. The distance they run is 480 meters.
We have entered the distances in figure 1.
Describe the figure 1 with measures of length
We have entered the distances in figure 1.
P4 -140: Can it be said that the animal, with all the hormones analyzed in physiological ranges, suffers stress? justify the answer.
We think that they do not suffer stress since all the values, although higher depending on the sample, are within the physiological range established for these breeds. And as we say in the discussion it can be a punctual acute stress due to the race of the animal.
P5 – 146: provide the normality test and ANOVA values, in each case that significant differences have been found: F -value and p -value, and also identify which values show the differences.
We have changed the statistical analysis in the material and methods section and introduced the following paragraph:
Analysis was performed using IBM SPSS Statistic 25 software (University Com-plutense of Madrid). The results were expressed as the means ± SE. The Kolmogorov–Smirnoff test was used to assess the goodness‐of‐fit distribution of the collected data. Var-iables studied were noted to be parametric, and an analysis of variance (ANOVA) was performed followed by Bonferroni post hoc test for the comparison between samples (sample 0, 1, 2) and between groups (salers and lidia calves). In all statistical analyses, the confidence level was 95%, and statistically significant differences were considered for p-values < 0.05.
In results section, we have changed the graphs and introduce the p-values resulted from the stadistical tests performed.
P6 – 174-177: It is not clear from the graphs where the significant differences are.
Thank you for appreciation. As significant differences were not clear, we changed the brackets in order to clarify the significances and also, we introduce the exact p-value obtained from the text as you requested previously.
P6 – 187: when talking about physiological ranges, only one article was referenced, they should look to see if there is a consensus on the ranges because they are sometimes very variable. It would be interesting to introduce more references of accepted physiological ranges.
Thank you for your comments, we have introduced numerous references in this regard.
P7- 202: end point missing.
Thank for your appreciation. We have introduced a final dot.
P7 – 206: It has been shown that the stress in cattle is more due to their immobilization for sample taking than to the puncture itself, change the argument.
We agree with your comment, but as we have also seen in the bibliography consulted, the puncture itself produces stress. We have added the following sentence in the text with its bibliographic citation:
and also minimizes the stress caused by the immobilization of the animal for blood col-lection [Szenci O, 2011]
P7- 239: As I have said, it is not true that calves are subjected to exercise on farms. justify the results from a genetic point of view. For this they have many articles to review on genetics and behavior of the lidia breed.
Thank you very much as we have said before, you are right, and for this reason we have changed the text as follows:
This may be due to the Lidia breed is characterized by its natural aggressiveness and re-sistance to traditional handling procedures. The performance of the Lidia bull during a bullfight could be compared to that of an athletic animal, due to the intense exercise per-formed in an unfamiliar and highly stressful environment [Escalera-Valente F, 2021].
P8 – 243: review more bibliography.
We have introduced numerous new bibliographic citations following your comments
P8 – 252: explain and describe what the temperament of these animals is like, reviewing scientific papers that corroborate it.
We have followed your suggestion and introduced the following paragraph:
The selection of the Lidia breed is based on its agonistic-aggressive behavior through a series of traits that classify its aggressiveness and fighting capacity. In addition, these traits show significant heritability, and thus can be considered suitable for genetic selection [Silva, Gonzalo and Cañón, 2006]. However, in other cattle breeds aggressiveness can be considered as an undesirable trait, it is likely that the selection process for aggressiveness in the Lidia breed has left genomic signatures [Eusebi, PG, 2021].
P8 – 265: the authors say that there is greater stress at some point, how have they measured it?
We have rewritten the sentence to make it clearer what we mean when we say that when there is an increase of cortisol there is at the same time a decrease in serotonin levels. For this we have put the following sentence:
which agrees with the results of this study, that when an increase in cortisol levels was found, at the same time, we find a decrease in serotonin levels.
The authors speak of a correlation in the concentrations of the hormones but no correlation analysis has been done, it would be interesting and obligatory to do so if they want to maintain the affirmations.
We have not measured the correlation we only refer to the quote from Bruschetta et al., 2010 that observed this negative correlation.
P8- 283: Is the test a sport or a livestock practice?
The contest is considered a sport by the Spanish Equestrian Federation.
It is necessary and interesting to discuss the different concentrations found in the different races, because these differences exist and to talk about how the sex variable can influence them. For all this, it is necessary to carry out a good bibliographical review on the subject since there are several publications on blood values in lidia cattle that the authors have not reviewed, they only refer to some thesis of the authors' research group.
Thank you for your comment: as you can see we have put new bibliography and we have read the works of Escalera Valente of 2013 and 2021 and we have included them in the text and bibliography. We have not talked about sex differences because in our case all the animals studied were females.

Reviewer 2 Report
Thank you for letting me review this manuscript. Unfortunately, in my opinion, it should be improved. Editing and formatting should be more accurate and several information is missing, especially in materials and methods.
In INTRODUCTION I would add something more about serotonin and dopamine and why it was decided to consider them in the study. Furthermore, hypotheses and predictions should be added.
MATERIALS AND METHODS are not precise and do not provide sufficiently detailed information (where was the saliva contained? What types of swabs were used? Were the animals used to saliva sampling?). Also, sampling timelines are vague and unspecified. When deciding to measure the hormones in saliva, the precise timing is extremely important and must be specified.
780 total samples were collected but how many from one group and how many from the other group?
IN THE RESULTS, the captions of the figures could state whether the bar indicates the standard deviation or the standard error.
In discussions and conclusions, there are very strong statements when in reality the quantification of hormones in saliva does not indicate the real level of welfare of the animals. Furthermore, the considered hormones vary not only in response to stress but also in response to physical exercise and/or may be indicative of the arousal and not of the valence of the animal's affective states. From the results obtained, you can only suppose something, but, in my opinion, your results are not about animal welfare, further evaluation should be required (e.g., behavioural) and I would repeat the hormonal measurement at least after 24 hours. Furthermore, I would also underline the limitations of the study.
Further comments are throughout the manuscript.

Author Response
Comments and Suggestions for Authors
Thank you for letting me review this manuscript. Unfortunately, in my opinion, it should be improved. Editing and formatting should be more accurate and several information is missing, especially in materials and methods.
Thank you for all your suggestions that have helped to improve the quality of the manuscript.
In INTRODUCTION I would add something more about serotonin and dopamine and why it was decided to consider them in the study. Furthermore, hypotheses and predictions should be added.
Thank you for your suggestion, we have made the required changes in the introduction section.
MATERIALS AND METHODS are not precise and do not provide sufficiently detailed information (where was the saliva contained? What types of swabs were used? Were the animals used to saliva sampling?). Also, sampling timelines are vague and unspecified. When deciding to measure the hormones in saliva, the precise timing is extremely important and must be specified.
Thank you for your appreciation. We have entered the required information in the text in the material and methods section.
780 total samples were collected but how many from one group and how many from the other group?
Three samples were collected per animal, 110 animals from the Salers group and 150 animals from the Lidia group, the total number of samples was 780.
IN THE RESULTS, the captions of the figures could state whether the bar indicates the standard deviation or the standard error.
Thank you for your suggestion, we have changed the graphics and figure legends.
In discussions and conclusions, there are very strong statements when in reality the quantification of hormones in saliva does not indicate the real level of welfare of the animals. Furthermore, the considered hormones vary not only in response to stress but also in response to physical exercise and/or may be indicative of the arousal and not of the valence of the animal's affective states. From the results obtained, you can only suppose something, but, in my opinion, your results are not about animal welfare, further evaluation should be required (e.g., behavioural) and I would repeat the hormonal measurement at least after 24 hours. Furthermore, I would also underline the limitations of the study.
Thank you for your appreciation. We have eliminated the phrase of animal welfare because you are right and with our results and without a behavioral test we cannot speak of animal welfare.
Regarding that the various hormones in response to exercise, that is something we say in the discussion with the values of cortisol and epinephrine.
Regarding the 24 hours after the event, we could not perform them (which we would have liked to) because the animals were transported to their farms of origin, and then we were going to introduce one more variable in the study which was the transport, which, as we know, causes a lot of stress to the animals. That is why we decided to take the picture two hours later, because after that time the animals were already being transported.
Further comments are throughout the manuscript.
Thank you very much we have taken into account all your considerations added in the text.
Regarding the comment: "were this range determined with the same methods employed by authors?" the concentrations of the studied hormones have been analyzed with the same enzymatic methods.

Round 2
Reviewer 1 Report
Please avoid using the word "fight" to name the Lidia breed or the "bullfighting" itself, is better "corrida".I still have serious doubts that Cossio and Cañabate was able to accurately measure the speed of the calf during the competition in 1988...
Lastly, there are more scientific articles on the lidia breed that they should use to reaffirm their assertions.
Author Response
Comments and Suggestions for Authors
Please avoid using the word "fight" to name the Lidia breed or the "bullfighting" itself, is better "corrida".
Thank you for your appreciations. We have changed in the text.
I still have serious doubts that Cossio and Cañabate was able to accurately measure the speed of the calf during the competition in 1988...
Thank you very much for your suggestion. We have searched in different bibliographic resources such as pubmed, web of science, etc.. And we have not found any citation about the speed of the fighting bull, we have only found the citation of Sinclair et al., 2016 which says that the speed of rodeo steers is about 20 Km/h. For this reason we have changed the speed to 20 - 30 Km/h.
Although in many informative pages we have found that the maximum speed of the bull in the running of the bulls of San Fermines from Pamplona, Spain, is 40 Km /h.
In the text we have put the speed from 20 to 30 Km/h and we have included the quote from Sinclair et al., 2016.
Lastly, there are more scientific articles on the lidia breed that they should use to reaffirm their assertions.
Thank you very much for your suggestion.
We have reviewed up to 30 articles on the fighting bull and in almost none of them we have found that study the hormones that we have studied.
The ones we have found are already introduced in the discussion and in the bibliography.

Reviewer 2 Report
The authors have done a good job. However, I believe that in its current form, there are still some shortcomings:
- I suggest removing the Spanish sentences from Figure 1
- The materials and methods section needs further details. For instance, I would include the average age of the animals, their weight, and possibly the Body Condition Score (BCS) if it was considered as a parameter.
- The description of the breeds you provided is entirely inadequate (please be more elaborate). Additionally, please provide more information about where the animals were sourced and the rearing conditions. What was the dietary regime followed by the subjects involved in the research?
- In Figure 2 and elsewhere, I would change 'sample 0, 1, and 2' to 'before, during, and after.'
- In the discussion section, please include practical applications of this research
- Additionally, the ethical committee's approval should be included.
Author Response
Comments and Suggestions for Authors
The authors have done a good job. However, I believe that in its current form, there are still some shortcomings:
Thank you very much for appreciating the work we have done in the previous review.
- I suggest removing the Spanish sentences from Figure 1
Thank you for your appreciations. We have changed the figure in the text.
- The materials and methods section needs further details. For instance, I would include the average age of the animals, their weight, and possibly the Body Condition Score (BCS) if it was considered as a parameter.
Thank you for your appreciations. We included in the text.
- The description of the breeds you provided is entirely inadequate (please be more elaborate). Additionally, please provide more information about where the animals were sourced and the rearing conditions. What was the dietary regime followed by the subjects involved in the research?
Thank you very much for your suggestion. We have changed the description of the study breeds in the text.
As it was a contest we did not have all these specifications, we do know the farms where they came from but due to data protection it is not possible to say them..
- In Figure 2 and elsewhere, I would change 'sample 0, 1, and 2' to 'before, during, and after.'
Thank you for your appreciations. We have changed in the text of the figure legends and we specify in material and methods.
- In the discussion section, please include practical applications of this research
Thank you for your appreciations. We included de practical aplications in the text.
- Additionally, the ethical committee's approval should be included.
The approval of the ethics committee is already included in the text. Thank you very much.

Round 3
Reviewer 2 Report
.
Author Response
Thank you very much for your suggestions